# Robust Feature-Sample Linear Discriminant Analysis for Brain Disorders Diagnosis

**Ehsan Adeli-Mosabbeb, Kim-Han Thung, Le An, Feng Shi, Dinggang Shen, for the ADNI**[*]
Department of Radiology and BRIC
University of North Carolina at Chapel Hill, NC, 27599, USA
{eadeli,khthung,le_an,fengshi,dgshen}@med.unc.edu

## Abstract

A wide spectrum of discriminative methods is increasingly used in diverse applications for classification or regression tasks. However, many existing discriminative methods assume that the input data is nearly noise-free, which limits their applications to solve real-world problems. Particularly for disease diagnosis, the data acquired by the neuroimaging devices are always prone to different sources of noise. Robust discriminative models are somewhat scarce and only a few attempts have been made to make them robust against noise or outliers. These methods focus on detecting either the sample-outliers or feature-noises. Moreover, they usually use unsupervised de-noising procedures, or separately de-noise the training and the testing data. All these factors may induce biases in the learning process, and thus limit its performance. In this paper, we propose a classification method based on the least-squares formulation of linear discriminant analysis, which simultaneously detects the sample-outliers and feature-noises. The proposed method operates under a semi-supervised setting, in which both labeled training and unlabeled testing data are incorporated to form the intrinsic geometry of the sample space. Therefore, the violating samples or feature values are identified as sample-outliers or feature-noises, respectively. We test our algorithm on one synthetic and two brain neurodegenerative databases (particularly for Parkinson's disease and Alzheimer's disease). The results demonstrate that our method outperforms all baseline and state-of-the-art methods, in terms of both accuracy and the area under the ROC curve.

## 1 Introduction

Discriminative methods pursue a direct mapping from the input to the output space for a classification or a regression task. As an example, linear discriminant analysis (LDA) aims to find the mapping that reduces the input dimensionality, while preserving the most class discriminatory information. Discriminative methods usually achieve good classification results compared to the generative models, when there are enough number of training samples. But they are limited when there are small number of labeled data, as well as when the data is noisy. Various efforts have been made to add robustness to these methods. For instance, [17] and [9] proposed robust Fisher/linear discriminant analysis methods, and [19] introduced a worst-case LDA, by minimizing the upper bound of the LDA cost function. These methods are all robust to *sample-outliers*. On the other hand, some methods were proposed to deal with the intra-sample-outliers (or *feature-noises*), such as [12, 15].

---

[*]Parts of the data used in preparation of this article were obtained from the Alzheimer's Disease Neuroimaging Initiative (ADNI) database (http://adni.loni.ucla.edu). The investigators within the ADNI contributed to the design and implementation of ADNI and/or provided data but did not participate in analysis or writing of this paper. A complete listing of ADNI investigators can be found at: http://adni.loni.ucla.edu/wp-content/uploads/howtoapply/ADNIAcknowledgementList.pdf.

As in many previous works, de-noising the training and the testing data are often conducted separately. This might induce a bias or inconsistency to the whole learning process. Besides, for many real-world applications, it is a cumbersome task to acquire enough training samples to perform a proper discriminative analysis. Hence, we propose to take advantage of the unlabeled testing data available, to build a more robust classifier. To this end, we introduce a semi-supervised discriminative classification model, which, unlike previous works, jointly estimates the noise model (both sample-outliers and feature-noises) on the whole labeled training and unlabeled testing data and simultaneously builds a discriminative model upon the de-noised training data.

In this paper, we introduce a novel classification model based on LDA, which is robust against both sample-outliers and feature-noises, and hence, here, it is called robust feature-sample linear discriminant analysis (RFS-LDA). LDA finds the mapping between the sample space and the label space through a linear transformation matrix, maximizing a so-called Fisher discriminant ratio [17]. In practice, the major drawback of the original LDA is the small sample size problem, which arises when the number of available training samples is less than the dimensionality of the feature space [18]. A reformulation of LDA based on the reduced-rank least-squares problem (known LS-LDA) [10] tackles this problem. LS-LDA finds the mapping $\boldsymbol{\beta} \in \mathbb{R}^{l \times d}$ by solving the following problem[1]:

$$\min_{\boldsymbol{\beta}} \|\mathbf{H}(\mathbf{Y}_{tr} - \boldsymbol{\beta}\mathbf{X}_{tr})\|_{\mathrm{F}}^2, \tag{1}$$

where $\mathbf{Y}_{tr} \in \mathbb{R}^{l \times N_{tr}}$ is a binary class label indicator matrix, for $l$ different classes (or labels), and $\mathbf{X}_{tr} \in \mathbb{R}^{d \times N_{tr}}$ is the matrix containing $N_{tr}$ $d$-dimensional training samples. $\mathbf{H}$ is a normalization factor defined as $\mathbf{H} = (\mathbf{Y}_{tr}\mathbf{Y}_{tr}^\top)^{-1/2}$ that compensates for the different number of samples in each class [10]. As a result, the mapping $\boldsymbol{\beta}$ is a reduced rank transformation matrix [10, 15], which could be used to project a test data $\mathbf{x}_{tst} \in \mathbb{R}^{d \times 1}$ onto a $l$ dimensional space. The class label could therefore be simply determined using a $k$-NN strategy.

To make LDA robust against noisy data, Fidler *et al.* [12] proposed to construct a basis, which contains complete discriminative information for classification. In the testing phase, the estimated basis identifies the outliers in samples (images in their case) and then is used to calculate the coefficients using a subsampling approach. On the other hand, Huang *et al.* [15] proposed a general formulation for robust regression (RR) and classification (robust LDA or RLDA). In the training stage, they de-noise the feature values using a strategy similar to robust principle component analysis (RPCA) [7] and build the above LS-LDA model using the de-noised data. In the testing stage, they de-noise the data by performing a locally compact representation of the testing samples from the de-noised training data. This separate de-noising procedure could not effectively form the underlying geometry of sample space to de-noise the data. Huang *et al.* [15] only account for feature-noise by imposing a sparse noise model constraint on the features matrix. On the other hand, the data fitting term in (1) is vulnerable to large sample-outliers. Recently, in robust statistics, it is found that $\ell_1$ loss functions are able to make more reliable estimations [2] than $\ell_2$ least-squares fitting functions. This has been adopted in many applications, including robust face recognition [28] and robust dictionary learning [22]. Reformulating the objective in (1), using this idea, would yield to this problem:

$$\min_{\boldsymbol{\beta}} \|\mathbf{H}(\mathbf{Y}_{tr} - \boldsymbol{\beta}\mathbf{X}_{tr})\|_1. \tag{2}$$

We incorporate this fitting function in our formulation to deal with the sample-outliers by iteratively re-weighting each single sample, while simultaneously de-noising the data from feature-noises. This is done through a semi-supervised setting to take advantage of all labeled and unlabeled data to build the structure of the sample space more robustly. Semi-supervised learning [8, 34] has long been of great interest in different fields, because it can make use of unlabeled or poorly labeled data. For instance, Joulin and Bach [16] introduced a convex relaxation and use the model in different semi-supervised learning scenarios. In another work, Cai *et al.* [5] proposed a semi-supervised discriminant analysis, where the separation between different classes is maximized using the labeled data points, while the unlabeled data points estimate the structure of the data. In contrast, we incorporate the unlabeled testing data to form the intrinsic geometry of the sample space and de-noise the data, whilst building the discriminative model.

$$\mathbf{X} = [\mathbf{X}_{tr}\ \mathbf{X}_{tst}] \in \mathbb{R}^{d \times N} \qquad \mathbf{D} = [\mathbf{D}_{tr}\ \mathbf{D}_{tst}] \in \mathbb{R}^{d \times N} \qquad \mathbf{E} \in \mathbb{R}^{d \times N}$$

Figure 1: Outline of the proposed method: The original data matrix, $\mathbf{X}$, is composed of both labeled training and unlabeled testing data. Our method decomposes this matrix to a de-noised data matrix, $\mathbf{D}$, and an error matrix, $\mathbf{E}$, to account for *feature-noises*. Simultaneously, we learn a mapping from the de-noised training samples in $\mathbf{D}$ ($\mathbf{D}_{tr}$) through a robust $\ell_1$ fitting function, dealing with the *sample-outliers*. The same learned mapping on the testing data, $\mathbf{D}_{tst}$, leads to the test labels.

We apply our method for the diagnosis of neurodegenerative brain disorders. The term *neurodegenerative disease* is an umbrella term for debilitating and incurable conditions related to progressive degeneration or death of the cells in the brain nervous system. Although neurodegenerative diseases manifest with diverse pathological features, the cellular level processes resemble similar structures. For instance, Parkinson's disease (PD) mainly affects the basal ganglia region and the substansia nigra sub-region of the brain, leading to decline in generation of a chemical messenger, dopamine. Lack of dopamine yields loss of ability to control body movements, along with some non-motor problems (*e.g.*, depression, anxiety) [35]. In Alzheimer's disease (AD), deposits of tiny protein plaques yield into brain damage and progressive loss of memory [26]. These diseases are often incurable and thus, early diagnosis and treatment are crucial to slow down the progression of the disease in its initial stages. In this study, we use two popular databases: PPMI and ADNI. The former aims at investigating PD and its related disorders, while the latter is designed for diagnosing AD and its prodormal stage, known as mild cognitive impairment (MCI).

**Contributions:** The contribution of this paper would therefore be multi-fold: (1) We propose an approach to deal with the sample-outliers and feature-noises simultaneously, and build a robust discriminative classification model. The sample-outliers are penalized through an $\ell_1$ fitting function, by re-weighing the samples based on their prediction power, while discarding the feature-noises. (2) Our proposed model operates under a semi-supervised setting, where the whole data (labeled training and unlabeled testing samples) are incorporated to build the intrinsic geometry of the sample space, which leads to better de-noising the data. (3) We further select the most discriminative features for the learning process through regularizing the weights matrix with an $\ell_1$ norm. This is specifically of great interest for the neurodegenerative disease diagnosis, where the features from different regions of the brain are extracted, but not all the regions are associated with a certain disease. Therefore, the most discriminative regions in the brain that utmost affect the disease would be identified, leading to a more reliable diagnosis model.

## 2   Robust Feature-Sample Linear Discriminant Analysis (RFS-LDA)

Let's assume we have $N_{tr}$ training and $N_{tst}$ testing samples, each with a $d$-dimensional feature vector, which leads to a set of $N = N_{tr} + N_{tst}$ total samples. Let $\mathbf{X} \in \mathbb{R}^{d \times N}$ denote the set of all samples (both training and testing), in which each column indicates a single sample, and $\mathbf{y}_i \in \mathbb{R}^{1 \times N}$ their corresponding $i^{\text{th}}$ labels. In general, with $l$ different labels, we can define $\mathbf{Y} \in \mathbb{R}^{l \times N}$. Thus, $\mathbf{X}$ and $\mathbf{Y}$ are composed by stacking up the training and testing data as: $\mathbf{X} = [\mathbf{X}_{tr}\ \mathbf{X}_{tst}]$ and $\mathbf{Y} = [\mathbf{Y}_{tr}\ \mathbf{Y}_{tst}]$. Our goal is to determine the labels of the test samples, $\mathbf{Y}_{tst} \in \mathbb{R}^{l \times N_{tst}}$.

**Formulation:** An illustration of the proposed method is depicted in Fig 1. First, all the samples (labeled or unlabeled) are arranged into a matrix, $\mathbf{X}$. We are interested in de-noising this matrix. Following [14, 21], this could be done by assuming that $\mathbf{X}$ can be spanned on a low-rank subspace and therefore should be rank-deficient. This assumption supports the fact that samples from same classes should be more correlated [14, 15]. Therefore, the original matrix $\mathbf{X}$ is decomposed into two

counterparts, $\mathbf{D}$ and $\mathbf{E}$, which represent the de-noised data matrix and the error matrix, respectively, similar to RPCA [7]. The de-noised data matrix shall hold the low-rank assumption and the error matrix is considered to be sparse. But, this process of de-noising does not incorporate the label information and is therefore unsupervised. Nevertheless, note that we also seek a mapping between the de-noised training samples and their respective labels. So, matrix $\mathbf{D}$ should be spanned on a low-rank subspace, which also leads to a good classification model of its sub-matrix, $\mathbf{D}_{tr}$.

To ensure the rank-deficiency of the matrix $\mathbf{D}$, like in many previous works [7, 14, 21], we approximate the rank function using the nuclear norm (the sum of the singular values of the matrix). The noise is modeled using the $\ell_1$ norm of the matrix, which ensures a sparse noise model on the feature values. Accordingly, the objective function for RFS-LDA under a semi-supervised setting would be:

$$\min_{\boldsymbol{\beta},\mathbf{D},\hat{\mathbf{D}},\mathbf{E}} \quad \frac{\eta}{2}\|\mathbf{H}(\mathbf{Y}_{tr} - \boldsymbol{\beta}\hat{\mathbf{D}})\|_1 + \|\mathbf{D}\|_* + \lambda_1\|\mathbf{E}\|_1 + \lambda_2\mathcal{R}(\boldsymbol{\beta}),$$
$$s.t. \quad \mathbf{D} = \mathbf{X} + \mathbf{E}, \hat{\mathbf{D}} = [\mathbf{D}_{tr}; \mathbf{1}^{\top}], \tag{3}$$

where the first term is the $\ell_1$ regression model introduced in (2). This term only operates on the de-noised training samples from matrix $\mathbf{D}$ with a row of all 1s is added to it, to ensure an appropriate linear classification model. The second and the third terms together with the first constraint are similar to the RPCA formulation [7]. They de-noise the labeled training and unlabeled testing data together. In combination with the first term, we ensure that the de-noised data also provides a favorable regression/classification model. The last term is a regularization on the learned mapping coefficients to ensure the coefficients do not get trivial or unexpectedly large values. The parameters $\eta$, $\lambda_1$ and $\lambda_2$ are constant regularization parameters, which are discussed in more details later.

The regularization on the coefficients could be posed as a simple norm of the $\boldsymbol{\beta}$ matrix. But, in many applications like ours (disease diagnosis) many of the features in the feature vectors are redundant. In practice, features from different brain regions are often extracted, but not all the regions contribute to a certain disease. Therefore, it is desirable to determine which features (regions) are the most relevant and the most discriminative to use. Following [11, 26, 28], we are looking for a sparse set of weights that ensures incorporating the least and the most discriminative features. We propose a regularization on the weights vector as a combination of the $\ell_1$ and Frobenius norms:

$$\mathcal{R}(\boldsymbol{\beta}) = \|\boldsymbol{\beta}\|_1 + \gamma\|\boldsymbol{\beta}\|_{\mathrm{F}}. \tag{4}$$

Evidently, the solution to the objective function in (3) is not easy to achieve, since the first term contains a quadratic term and minimization of the $\ell_1$ fitting function is not straightforward (because of its indifferentiability). To this end, we formalize the solution with a similar strategy as in iteratively re-weighted least squares (IRLS) [2]. The $\ell_1$ minimization problem is approximated by a conventional $\ell_2$ least-squares, in which each of the samples in the $\hat{\mathbf{D}}$ matrix are weighted with the reverse of their regression residual. Therefore the new problem would be:

$$\min_{\boldsymbol{\beta},\mathbf{D},\hat{\mathbf{D}},\mathbf{E}} \quad \frac{\eta}{2}\|\mathbf{H}(\mathbf{Y}_{tr} - \boldsymbol{\beta}\hat{\mathbf{D}})\hat{\boldsymbol{\alpha}}\|_{\mathrm{F}}^2 + \|\mathbf{D}\|_* + \lambda_1\|\mathbf{E}\|_1 + \lambda_2\mathcal{R}(\boldsymbol{\beta}),$$
$$s.t. \quad \mathbf{D} = \mathbf{X} + \mathbf{E}, \hat{\mathbf{D}} = [\mathbf{D}_{tr}; \mathbf{1}^{\top}]. \tag{5}$$

where $\hat{\boldsymbol{\alpha}}$ is a diagonal matrix, the $i^{\mathrm{th}}$ diagonal element of which is the $i^{\mathrm{th}}$ sample's weight:

$$\hat{\boldsymbol{\alpha}}_{ii} = \mathbf{1}/\sqrt{(\mathbf{y}_i - \boldsymbol{\beta}\hat{\mathbf{d}}_i)^2 + \delta}, \ \forall \ i,j \in \{0,\ldots,N_{tr}\}, i \neq j, \hat{\boldsymbol{\alpha}}_{ij} = 0, \tag{6}$$

where $\delta$ is a very small positive number (equal to $0.0001$ in our experiments). In the next subsection, we introduce an algorithm to solve this optimization problem.

Our work is closely related to the RR and RLDA formulations in [15], where the authors impose a low-rank assumption on the training data feature values and an $\ell_1$ assumption on the noise model. The discriminant model is learned similar to LS-LDA, as illustrated in (1), while a sample-weighting strategy is employed to achieve a more robust model. On the other hand, our model operates under a semi-supervised learning setting, where both the labeled training and the unlabeled testing samples are de-noised simultaneously. Therefore, the geometry of the sample space is better modeled on the low-dimensional subspace, by interweaving both labeled training and unlabeled testing data. In addition, our model further selects the most discriminative features to learn the regression/classification model, by regularizing the mapping weights vector and enforcing an sparsity condition on them.

**Algorithm 1** RFS-LDA optimization algorithm.

---

**Input:** $\mathbf{X} = [\mathbf{X}_{tr}\ \mathbf{X}_{tst}], \mathbf{Y}_{tr}$, parameters $\eta, \lambda_1, \lambda_2, \rho$ and $\gamma$.

**Initialization**: $\mathbf{D}^0 = [\mathbf{X}_{tr}\ \mathbf{X}_{tst}], \hat{\mathbf{D}}^0 = [\mathbf{X}_{tr}; \mathbf{1}^\top], \boldsymbol{\beta}^0 = \mathbf{Y}_{tr}(\hat{\mathbf{D}}^0)^\top(\hat{\mathbf{D}}^0(\hat{\mathbf{D}}^0)^\top + \gamma\mathbf{I}), \mathbf{E}^0 = \mathbf{0}, \mathscr{L}_1^0 = \mathbf{X}/\|\mathbf{X}\|_2, \mathscr{L}_2^0 = \mathbf{x}_{tr}/\|\mathbf{X}_{tr}\|_2, \mathscr{L}_3^0 = \boldsymbol{\beta}^0/\|\boldsymbol{\beta}^0\|_2, \mu_1 = \frac{dN}{4}\|\mathbf{X}\|_1, \mu_2 = \frac{dN_{tr}}{4}\|\mathbf{X}_{tr}\|_1, \mu_3 = \frac{dc}{4}\|\boldsymbol{\beta}^0\|_1$.

1: $k \leftarrow 0$
2: **repeat**                                                      ▷ Main optimization loop
3:        $t \leftarrow 0, \hat{\boldsymbol{\beta}}^0 = \boldsymbol{\beta}^k$                                                        ▷ Update $\boldsymbol{\beta}$
4:        **repeat**
5:              $\forall\, i, j \in \{0, \ldots, N_{tr} - 1\}, i \neq j, \hat{\boldsymbol{\alpha}}_{ij} \leftarrow 0$ and $\hat{\boldsymbol{\alpha}}_{ii} \leftarrow {}^1\!/\sqrt{(\mathbf{y}_i^k - \hat{\boldsymbol{\beta}}^t \hat{\mathbf{d}}_i^k)^2 + 0.0001}$
6:              $\hat{\boldsymbol{\beta}}^{t+1} \leftarrow \big(\mathbf{Y}_{tr}\hat{\boldsymbol{\alpha}}\hat{\boldsymbol{\alpha}}^\top(\hat{\mathbf{D}}^k)^\top + \mu_3(\mathbf{B}^k - \mathscr{L}_3^k)\big)\big(\hat{\mathbf{D}}^k\hat{\boldsymbol{\alpha}}\hat{\boldsymbol{\alpha}}^\top(\hat{\mathbf{D}}^k)^\top + \gamma\mathbf{I}\big), t \leftarrow t + 1$
7:        **until** $\|\hat{\boldsymbol{\beta}}^{t-1} - \hat{\boldsymbol{\beta}}^t\|_{\mathrm{F}}/(\|\hat{\boldsymbol{\beta}}^{t-1}\|_{\mathrm{F}} \times \|\hat{\boldsymbol{\beta}}^t\|_{\mathrm{F}}) < 0.001$ or $t > 100$
8:        $\boldsymbol{\beta}^{k+1} \leftarrow \hat{\boldsymbol{\beta}}^t$.
9:        $\hat{\mathbf{D}}^{k+1} \leftarrow \big(\eta\hat{\boldsymbol{\alpha}}^\top(\boldsymbol{\beta}^{k+1})^\top\boldsymbol{\beta}^{k+1}\hat{\boldsymbol{\alpha}} + \mu_2^k\mathbf{I}\big)^{-1}\big(\eta\hat{\boldsymbol{\alpha}}^\top(\boldsymbol{\beta}^{k+1})^\top\mathbf{Y}_{tr} - \mathscr{L}_2^k + \mu_2^k[\mathbf{D}_{tr}^k; \mathbf{1}^\top]\big)$    ▷ Update $\hat{\mathbf{D}}$
10:       $\mathbf{D}^{k+1} \leftarrow \mathcal{D}_{1/(\mu_1^k + \mu_2^k)}\big(\mathscr{L}_1^k + \mu_1^k(\mathbf{X} - \mathbf{E}^k) + \big[[\mathscr{L}_2^k + \mu_2^k\hat{\mathbf{D}}^{k+1}]_{(1:N_{tr},:)}\ \mathbf{0}\big]\big)$    ▷ Update $\mathbf{D}$
11:       $\mathbf{E}^{k+1} \leftarrow \mathcal{S}_{\lambda_1/\mu_1^k}(\mathbf{X} - \mathbf{D}^{k+1} + \mathscr{L}_1^k/\mu_1^k)$                                ▷ Update $\mathbf{E}$
12:       $\mathbf{B}^{k+1} \leftarrow \mathcal{S}_{\lambda_2/\mu_3^k}(\boldsymbol{\beta}^{k+1} + \mathscr{L}_3^k)$                                        ▷ Update $\mathbf{B}$
13:       $\mathscr{L}_1^{k+1} \leftarrow \mathscr{L}_1^k + \mu_1^k(\mathbf{X} - \mathbf{D}^{k+1} - \mathbf{E}^{k+1})$             ▷ Update multipliers and parameters
14:       $\mathscr{L}_2^{k+1} \leftarrow \mathscr{L}_2^k + \mu_2^k(\hat{\mathbf{D}} - [\mathbf{D}_{tr}^{k+1}; \mathbf{1}^\top]), \mathscr{L}_3^{k+1} \leftarrow \mathscr{L}_3^k + \mu_3^k(\boldsymbol{\beta} - \mathbf{B})$
15:       $\mu_1^{k+1} \leftarrow \min(\rho\mu_1^k, 10^9), \mu_2^{k+1} \leftarrow \min(\rho\mu_2^k, 10^9), \mu_3^{k+1} \leftarrow \min(\rho\mu_3^k, 10^9)$
16:       $k \leftarrow k + 1$
17: **until** $\|\mathbf{X} - \mathbf{D}^k - \mathbf{E}^k\|_{\mathrm{F}}/\|\mathbf{X}\|_{\mathrm{F}} < 10^{-8}$ and $\|\hat{\mathbf{D}}^k - [\mathbf{D}_{tr}^k; \mathbf{1}^\top]\|_{\mathrm{F}}/\|\hat{\mathbf{D}}^k\|_{\mathrm{F}} < 10^{-8}$ and $\|\boldsymbol{\beta}^k - \mathbf{B}^k\|_{\mathrm{F}}/\|\boldsymbol{\beta}^k\|_{\mathrm{F}} < 10^{-8}$
      **Output:** $\boldsymbol{\beta}, \mathbf{D}, \mathbf{E}$ and $\mathbf{Y}_{tst} = \boldsymbol{\beta}\mathbf{X}_{tst}$.

---

**Optimization:** Problem (5) could be efficiently solved using the augmented Lagrangian multipliers (ALM) approach. Hence, we introduce the Lagrangian multipliers, $\mathscr{L}_1 \in \mathbb{R}^{d \times N}, \mathscr{L}_2 \in \mathbb{R}^{(d+1) \times N_{tr}}$ and $\mathscr{L}_3 \in \mathbb{R}^{l \times (d+1)}$, an auxiliary variable, $\mathbf{B} \in \mathbb{R}^{l \times (d+1)}$, and write the Lagrangian function as:

$$
\begin{aligned}
\mathcal{L}(\boldsymbol{\beta}, \mathbf{B}, \mathbf{D}, \hat{\mathbf{D}}, \mathbf{E}) = &\frac{\eta}{2}\|\mathbf{H}(\mathbf{Y}_{tr} - \boldsymbol{\beta}\hat{\mathbf{D}})\hat{\boldsymbol{\alpha}}\|_{\mathrm{F}}^2 + \|\mathbf{D}\|_* + \lambda_1\|\mathbf{E}\|_1 + \lambda_2(\|\mathbf{B}\|_1 + \gamma\|\boldsymbol{\beta}\|_{\mathrm{F}}) \\
&+ \langle\mathscr{L}_1, \mathbf{X} - \mathbf{D} - \mathbf{E}\rangle + \frac{\mu_1}{2}\|\mathbf{X} - \mathbf{D} - \mathbf{E}\|_{\mathrm{F}}^2 + \langle\mathscr{L}_2, \hat{\mathbf{D}} - [\mathbf{D}_{tr}; \mathbf{1}^\top]\rangle \quad (7) \\
&+ \frac{\mu_2}{2}\|\hat{\mathbf{D}} - [\mathbf{D}_{tr}; \mathbf{1}^\top]\|_{\mathrm{F}}^2 + \langle\mathscr{L}_3, \boldsymbol{\beta} - \mathbf{B}\rangle + \frac{\mu_3}{2}\|\boldsymbol{\beta} - \mathbf{B}\|_{\mathrm{F}}^2,
\end{aligned}
$$

where $\mu_1, \mu_2$ and $\mu_3$ are penalty parameters. There are five variables ($\boldsymbol{\beta}, \mathbf{B}, \mathbf{D}, \hat{\mathbf{D}}$ and $\mathbf{E}$) contributing to the problem. We alternatively optimize for each variable, while fixing the others. Except for the matrix $\boldsymbol{\beta}$, all the variables have straightforward or closed-form solutions. $\boldsymbol{\beta}$ is calculated through IRLS [2], by iteratively calculating the weights in $\hat{\boldsymbol{\alpha}}$ and solving the conventional least-squares problem, until convergence.

The detailed optimization steps are given in Algorithm 1. The normalization factor $\mathbf{H}$ is omitted in this algorithm, for easier readability. In this algorithm, $\mathbf{I}$ is the identity matrix and the operators $\mathcal{D}_\tau(.)$ and $\mathcal{S}_\kappa(.)$ are defined in the following. $\mathcal{D}_\tau(\mathbf{A}) = \mathbf{U}\mathcal{D}_\tau(\boldsymbol{\Sigma})\mathbf{V}^*$ applies singular value thresholding algorithm [6] on the intermediate matrix $\boldsymbol{\Sigma}$, as $\mathcal{D}_\tau(\boldsymbol{\Sigma}) = \mathrm{diag}(\{(\sigma_i - \tau)_+\})$, where $\mathbf{U}\boldsymbol{\Sigma}\mathbf{V}^*$ is the singular values decomposition (SVD) of $\mathbf{A}$ and $\sigma_i$s are the singular values. Additionally, $\mathcal{S}_\kappa(a) = (a - \kappa)_+ - (-a - \kappa)_+$ is the soft thresholding operator or the proximal operator for the $\ell_1$ norm [3]. Note that $s_+$ is the positive part of $s$, defined as $s_+ = \max(0, s)$.

**Algorithm analysis:** The solution for each of the matrices $\mathbf{B}, \mathbf{D}, \hat{\mathbf{D}}, \mathbf{E}$ is a convex function, while all the other variables are fixed. For $\boldsymbol{\beta}$, the solution is achieved via the IRLS approach, in an iterative manner. Both the $\ell_1$ fitting function and the approximated re-weighted least-squares functions are convex. We only need to ensure that the minimization of the latter is numerically better tractable than the minimization of the former. This is discussed in depth and the convergence is proved in [2].

To estimate the computational complexity of the algorithm, we need to investigate the complexity of the sub-procedures of the algorithm. The two most computationally expensive steps in the loop are the iterative update of $\boldsymbol{\beta}$ (Algorithm 1, Steps 4-7) and the SVT operation (Algorithm 1, Step 10). The former includes solving a least-squares iteratively, which is $O(d^2N)$ in each iteration and the latter has the SVD operation as the most computational intensive operation, which is of $O(d^2N + N^3)$.

By considering the maximum number of iterations for the first sub-procedure equal to $t_{max} = 100$, the overall computational complexity of the algorithm in each iteration would be $O(100d^2N + N^3)$. The number of iterations of the whole algorithm until convergence is dependent on the choice of $\{\mu\}$s. If $\mu$ penalty parameters are increasing smoothly in each iteration (as in Step 15, Algorithm 1), the overall algorithm would be Q-linearly convergent. A reasonable choice for the sequence of all $\{\mu\}$s yields in a decrease in the number of required SVD operations [1, 21].

## 3    Experiments

We compare our method with several baseline and state-of-the-art methods in three different scenarios. The first experiment is on synthetic data, which highlights how the proposed method is robust against sample-outliers or feature-noises, separately or when they occur at the same time. The next two experiments are conducted for neurodegenerative brain disorders diagnosis. We use two popular databases, one for Parkinson's disease (PD) and the other for Alzheimer's disease (AD).

We compare our results with different baseline methods, including: Conventional LS-LDA [10], RLDA [15], RPCA on the $\mathbf{X}$ matrix separately to de-noise and then LS-LDA for the classification (denoted as RPCA+LS-LDA) [15], linear support vector machines (SVM), and sparse feature selection with SVM (SFS+SVM) or with RLDA (SFS+RLDA). Except for RPCA+LDA, the other methods in comparison do not incorporate the testing data. In order to have a fair set of comparisons, we also compare against the transductive matrix completion (MC) approach [14]. Additionally, to also evaluate the effect of the regularization on matrix $\boldsymbol{\beta}$, we report results for RFS-LDA when regularized by only $\gamma\|\boldsymbol{\beta}\|_F$ (denoted as RFS-LDA*), instead of the term introduced in (4). Moreover, we also train our proposed RFS-LDA in a fully supervised setting, *i.e.*, not involving any testing data in the training process, to show the effect of the established semi-supervised learning framework in our proposed method. This is simply done by replacing variable $\mathbf{X}$ in (3) with $\mathbf{X}_{tr}$ and solving the problem correspondingly. This method, referred to as S-RFS-LDA, only uses the training data to form the geometry of the sample space and, therefore, only cleans the training feature-noises.

For the choice of parameters, the best parameters are selected through an inner 10-fold cross validation on the training data, for all the competing methods. For the proposed method, the parameters are set with a same strategy as in [15]: $\lambda_1 = \Lambda_1/(\sqrt{\min(d,N)})$, $\lambda_2 = \Lambda_2/\sqrt{d}$, $\eta^k = \Lambda_3\|\mathbf{X}\|_*/\|\mathbf{Y}_{tr} - \boldsymbol{\beta}^k\hat{\mathbf{D}}^k\|_F^2$, and $\rho$ (controlling the $\{\mu\}$s in the algorithm) is set to $1.01$. We have set $\Lambda_1, \Lambda_2, \Lambda_3$ and $\gamma$ through inner cross validation, and found that all set to $1$ yields to reasonable results across all datasets.

**Synthetic Data:** We construct two independent 100-dimensional subspaces, with bases $\mathbf{U}_1$ and $\mathbf{U}_2$ (same as described in [21]). $\mathbf{U}_1 \in \mathbb{R}^{100 \times 100}$ is a random orthogonal matrix and $\mathbf{U}_2 = \mathbf{T}\mathbf{U}_1$, in which $\mathbf{T}$ is a random rotation matrix. Then, 500 vectors are sampled from each subspace through $\mathbf{X}_i = \mathbf{U}_i\mathbf{Q}_i, i = \{1,2\}$, with $\mathbf{Q}_i$, a $100 \times 500$ matrix, independent and identically distributed (*i.i.d.*) from $\mathcal{N}(0,1)$. This leads to a binary classification problem. We gradually add additional noisy samples and features to the data, drawn *i.i.d* from $\mathcal{N}(0,1)$, and evaluate our proposed method. The accuracy means and standard deviations of three different runs are illustrated in Fig. 2. This experiment is conducted under three settings: (1) First, we analyze the behavior of the method against gradually added noise to some of the features (feature-noises), illustrated in Fig. 2a. (2) We randomly add some noisy samples to the aforementioned noise-free samples and evaluate the methods in the sole presence of sample-outliers. Results are depicted in Fig. 2b. (3) Finally, we simultaneously add noisy features and samples. Fig. 2c shows the mean±std accuracy as a function of the additional number of noisy features and samples. Note that all the reported results are obtained through 10-fold cross-validation. As can be seen, our method is able to select a better subset of features and samples and achieve superior results compared to RLDA and conventional LS-LDA approaches. Furthermore, our method behaves more robust against the increase in the noise factor.

**Brain neurodegenrative disease diagnosis databases:** The first set of data used in this paper is obtained from the Parkinson's progression markers initiative (PPMI) database[2] [23]. PPMI is the first substantial study for identifying the PD progression biomarkers to advance the understanding of the disease. In this research, we use the MRI data acquired by the PPMI study, in which a T1-weighted, 3D sequence (*e.g.*, MPRAGE or SPGR) is acquired for each subject using 3T SIEMENS MAGNETOM TrioTim syngo scanners. We use subjects scanned using MPRAGE sequence to

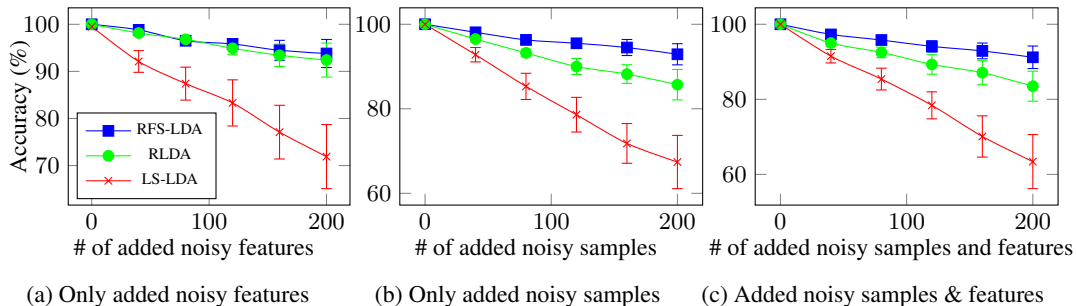

<div align="center">(a) Only added noisy features     (b) Only added noisy samples     (c) Added noisy samples & features</div>

Figure 2: Results comparisons on synthetic data, for three different runs (mean±std).

Table 1: The accuracy (ACC) and area under ROC curve (AUC) of the PD/NC classification on PPMI database, compared to the baseline methods.

| | | | | Method | | | | | | |
|---|---|---|---|---|---|---|---|---|---|---|
| | RFS-LDA | RFS-LDA$^*$ | S-RFS-LDA | RLDA | SFS+RLDA | RPCA+LS-LDA | LS-LDA | SVM | SFS+SVM | MC |
| ACC | **84.1** | 78.3 | 75.8 | 71.0 | 73.4 | 59.4 | 56.6 | 55.2 | 61.5 | 61.5 |
| AUC | **0.87** | 0.81 | 0.80 | 0.79 | 0.80 | 0.64 | 0.59 | 0.56 | 0.59 | 68.8 |

minimize the effect of different scanning protocols. The T1-weighted images were acquired for 176 sagittal slices with the following parameters: repetition time = 2300 ms, echo time = 2.98 ms, flip angle = $9°$, and voxel size = $1 \times 1 \times 1$ mm$^3$. All the MR images were preprocessed by skull stripping [29], cerebellum removal, and then segmented into white matter (WM), gray matter (GM), and cerebrospinal fluid (CSF) tissues [20]. The anatomical automatic labeling atlas [27], parcelled with 90 predefined regions of interest (ROI), was registered using HAMMER[3] [25, 30] to each subject's native space. We further added 8 more ROIs in basal ganglia and brainstem regions, which are clinically important ROIs for PD. We then computed WM, GM and CSF tissue volumes in each of the 98 ROIs as features. 56 PD and 56 normal control (NC) subjects are used in our experiments.

The second dataset is from Alzheimer's disease neuroimaging initiative (ADNI) study[4], including MRI and FDG-PET data. For this experiment, we used 93 AD patients, 202 MCI patients and 101 NC subjects. To process the data, same tools employed in [29] and [32] are used, including spatial distortion, skull-stripping, and cerebellum removal. The FSL package [33] was used to segment each MR image into three different tissues, *i.e.*, GM, WM, and CSF. Then, 93 ROIs are parcelled for each subject [25] with atlas warping. The volume of GM tissue in each ROI was calculated as the image feature. For FDG-PET images, a rigid transformation was employed to align it to the corresponding MR image and the mean intensity of each ROI was calculated as the feature. All these features were further normalized in a similar way, as in [32].

**Results:** The first experiment is set up on the PPMI database. Table 1 shows the diagnosis accuracy of the proposed technique (RFS-LDA) in comparisons with different baseline and state-of-the-art methods, using a 10-fold cross-validation strategy. As can be seen, the proposed method outperforms all others. This could be because our method deals with both feature-noises and sample-outliers. Note that, subjects and their corresponding feature vectors extracted from MRI data are quite prone to noise, because of many possible sources of noise (*e.g.* the patient's body movements, RF emission due to thermal motion, overall MR scanner measurement chain, or preprocessing artifacts). Therefore, some samples might not be useful (sample-outliers) and some might be contaminated by some amounts of noise (feature-noises). Our method deals with both types and achieves good results.

The goal for the experiments on ADNI database is to discriminate both MCI and AD patients from NC subjects, separately. Therefore, NC subjects form our negative class, while the positive class is defined as AD in one experiment and MCI in the other. The diagnosis results of the AD *vs*. NC and MCI *vs*. NC experiments are reported in Tables 2. As it could be seen, in comparisons with the state-of-the-art, our method achieves good results in terms of both accuracy and the area under curve. This is because we successfully discard the sample-outliers and detect the feature-noises.

Table 2: The accuracy (ACC) and the area under ROC curve (AUC) of the Alzheimer's disease classification on ADNI database, compared to the baseline methods.

| | | Method | | | | | | | | | |
|---|---|---|---|---|---|---|---|---|---|---|---|
| | | RFS-LDA | RFS-LDA* | S-RFS-LDA | RLDA | SFS+RLDA | RPCA+LS-LDA | LS-LDA | SVM | SFS+SVM | MC |
| AD/NC | ACC | **91.8** | 89.1 | 86.3 | 88.7 | 90.1 | 87.6 | 70.9 | 72.1 | 76.3 | 78.2 |
| | AUC | **0.98** | 0.96 | 0.95 | 0.96 | **0.98** | 0.93 | 0.81 | 0.80 | 0.83 | 0.82 |
| MCI/NC | ACC | **89.8** | 85.6 | 84.5 | 85.0 | 88.1 | 84.5 | 68.9 | 70.1 | 76.1 | 74.3 |
| | AUC | **0.93** | 0.90 | 0.90 | 0.87 | 0.92 | 0.87 | 0.75 | 0.79 | 0.80 | 0.78 |

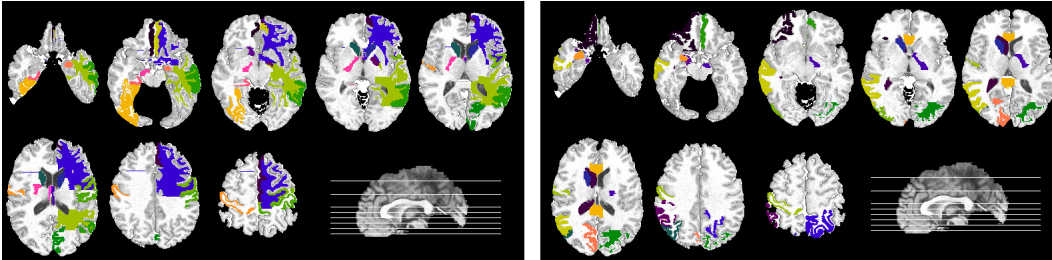

Figure 3: The top selected ROIs for AD *vs*. NC (left) and MCI *vs*. NC (right) classification problems.

**Discussions:** In medical imaging applications, many sources of noise (*e.g.* patient's movement, radiations and limitation of imaging devices, preprocessing artifacts) contribute to the acquired data [13], and therefore methods that deal with noise and outliers are of great interest. Our method enjoys from a single optimization objective that can simultaneously suppress sample-outliers and feature-noises, which compared to the competing methods, exhibits a good performance. One of the interesting functions of the proposed method is the regularization on the mapping coefficients with the $\ell_1$ norm, which would select a compact set of features to contribute to the learned mapping. The magnitude of the coefficients would show the level of contribution of that specific feature to the learned model. In our application, the features from the whole brain regions are extracted, but only a small number of regions are associated with the disease (*e.g*., AD, MCI or PD). Using this strategy, we can determine which brain regions are highly associated with a certain disease.

Fig. 3 shows the top regions selected by our algorithm in AD *vs*. NC and MCI *vs*. NC classification scenarios. These regions, including middle temporal gyrus, medial front-orbital gyrus, postcentral gyrus, caudate nucleus, cuneus, and amygdala have been reported to be associated with AD and MCI in the literature [24, 26]. The figures show the union of regions selected for both MRI and FDG-PET features. The most frequently used regions for the PD/NC experiment are the substantial nigra (left and right), putamen (right), middle frontal gyrus (right), superior temporal gyrus (left), which are also consistent with the literature [4, 31]. This selection of brain regions could be further incorporated for future clinical analysis.

The semi-supervised setting of the proposed method is also of great interest in the diagnosis of patients. When new patients first arrive and are to be diagnosed, the previous set of the patients with no certain diagnosis so far (not labeled yet), could still be used to build a more reliable classifier. In other words, the current testing samples could contribute the diagnosis of future subjects, as unlabeled samples.

## 4 Conclusion

In this paper, we proposed an approach for discriminative classification, which is robust against both sample-outliers and feature-noises. Our method enjoys a semi-supervised setting, where all the labeled training and the unlabeled testing data are used to detect outliers and are de-noised, simultaneously. We have applied our method to the interesting problem of neurodegenerative brain disease diagnosis and directly applied it for the diagnosis of Parkinson's and Alzheimer's diseases. The results show that our method outperforms all competing methods. As a direction for the future work, one can develop a multi-task learning reformulation of the proposed method to incorporate multiple modalities for the subjects, or extend the method for the incomplete data case.

## Footnotes

[1]Bold capital letters denote matrices (*e.g.*, $\mathbf{D}$). All non-bold letters denote scalar variables. $d_{ij}$ is the scalar in the row $i$ and column $j$ of $\mathbf{D}$. $\langle \mathbf{d}_1, \mathbf{d}_2 \rangle$ denotes the inner product between $\mathbf{d}_1$ and $\mathbf{d}_2$. $\|\mathbf{d}\|_2^2$ and $\|\mathbf{d}\|_1$ represent the squared Euclidean Norm and the $\ell_1$ norm of $\mathbf{d}$, respectively. $\|\mathbf{D}\|_{\mathrm{F}}^2 = \mathrm{tr}(\mathbf{D}^\top \mathbf{D}) = \sum_{ij} d_{ij}$ and $\|\mathbf{D}\|_*$ designate the squared Frobenius Norm and the nuclear norm (sum of singular values) of $\mathbf{D}$, respectively.

[2]http://www.ppmi-info.org/data

[3]Could be downloaded at http://www.nitrc.org/projects/hammerwml

[4]http://www.loni.ucla.edu/ADNI

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
