[Reviews · NeurIPS 2015]

Submitted by Assigned_Reviewer_1

This paper presents an robust and semisupervised approach to learn a Linear Discriminant Analysis (LDA) classifier, that can deal with both outlier samples and noisy features. Evaluations on synthetic and MR-images classification for brain disorder diagnostic application show the superiority of the method as compared to alternative robust algorithms. The formulation is rigorous and relevant leading to an algorithm that seem overall to be a nice contribution to machine learning and MR image-based diagnosis. The paper is also clear and nicely written. I nonetheless have a few comments and suggestions about it:

- If I am not wrong (I may be), the proposed optimization algorithm only finds a local optimum, but may not find the overall best solution, is that right? If so please mention it. Also, how sensitive is the algorithm to the initial conditions (beta0, D0, etc.)?

- Regarding the synthetic data, I would have liked to obtain more information on their distribution, how the noise was defined, etc. So far we know it is increasingly noisy data, but not much more. Providing more information would help to reproduce the results.

- Regarding the targeted application, i.e., brain disorder diagnosis based on MR image classification, what are the state-of-the-art methods currently used? Why not comparing the proposed approach to these state-of-the-art method? For instance, can non-linear classifiers be useful for this application domain? (only linear classifiers were used here)

- ALthough a distinctive feature of the proposed approach is the semi-supervised setting, it is not clear how this really helps. It would have been nice to compare the proposed approach with and without unlabelled data, to see whether using semi-supervised learning does improve the results and if so, how much.

minor points:

- why focusing on LDA?

- in the introduction, the description of paper [11] is not so clear. In particular, what does mean "a basis with complete discriminative information", and what are the coefficients mentioned for the testing phase?

- in equation 2, there is an extra power 1

- when mentioning semi-supervised learning, why mentioning [14], [5] and [1] more than other works? Some review references may be relevant here:

Xiaojin Zhu. Semi-supervised learning literature survey. Technical Report 1530, Department of Computer Sciences, University of Wisconsin, Madison, 2005

Semi-supervised learning O Chapelle, B Schoelkopf, A Zien - MIT press - 2006
Summary: This paper presents a rigorous and relevant formulation to learn a robust and semisupervised Linear Discriminant Analysis (LDA) classifier, that can deal with both outlier samples and noisy features. Evaluations on multiple data sets suggests that the methode indeed improves on the state of the art and is therefore a valuable contribution.

Submitted by Assigned_Reviewer_2

This work was vaguely interesting as it proposed handling noisy features and labels within a semi-supervised pattern recognition framework.

In medicine, a proportion of labels are often wrong (eg a diagnosis of Alzheimer's disease is often incorrect).

In terms of originality, there are many papers on the subject of improving classification accuracy using the ADNI (Alzheimer's disease, AD) dataset, although not as many using the PPMI (Parkinson's disease, PD) database.

It would perhaps have been interesting to see if the proposed method generalises to other sorts of data beyond diagnoses using MRI.

It could also be interesting to assess the behaviour of the approach if some of the training data are deliberately mislabelled.

This could be future work though.

This manuscript builds on the Huang et al paper on Robust regression, which I have not read. The approach involves various settings, which were determined via a nested cross-validation procedure.

This should mean that the estimated classification accuracies are relatively unbiased.

For the PD dataset, there were 56 normal controls and 56 patients.

Classification accuracy was 84.1%, which was higher than for the other methods.

The dataset was relatively small, so it's difficult to say whether this was a statistical fluke. For the AD dataset, the proposed method also appeared to perform best, although the accuracy was not so much better than for the other methods.

This could easily have been down to chance.

Accuracies are not really high enough for clinical applications, but this may be down to incorrect diagnoses of patients involved.

If the method had been probabilistic, then prediction could perhaps be combined with other sources of information, but this is not the case here.

There is not even any mention of sensitivity and specificity, which could be used to adjust for the fact that in the real world, the prior probability of having a disease is not 0.5 (or whatever the proportions are in the training data).

The manuscript uses the ADNI dataset, but does not appear to comply with their data use agreement.

I do not personally approve of the requirements imposed by ADNI, and I do not know if NIPS has an editorial policy on this sort of thing.

I do not see how equation 1 gets around the N > d problem, as there is no regularization involved.

I do not understand why the authors propose the use of k-NN to determine class labels after having performed LDA.

Figure 1 is not needed and does not add anything to the work.

In the "Contributions", the authors say they "build a robust discriminative classification model".

My understanding of LDA is that it is fundamentally generative, which models two Gaussian distributions that share their covariance matrices.

The framework here seems to be discriminative though, treating classification as a regression problem.

In equations 2 and 3, is the 1 superscript needed in $|| H (Y_{tr} - \beta X_{tr}) ||_1^1$ ?

On page 4, the paper says that the first term of equation 3 contains a quadratic optimization step.

This does not make sense, as the equation does not contain steps.

The first term is an l_1 norm. The step from equation 3 to equation 5 needs some more rigorous justification.

It currently seems like an ad hoc procedure.

On page 8, the manuscript says "The magnitude of the coefficients would show the significance of contribution for that specific feature".

This is not a statistical significance, and it is not clinical significance.

Perhaps refer to the "relevance", rather than "significance".
Summary: Authors propose a robust semi-supervised regression method, which they apply to classification of diseases from features extracted from medical images.

Robustness is incorporated into the approach through penalising various norms of the matrices involved.

Submitted by Assigned_Reviewer_3

The paper 462, entitled 'Semi-Supervised Robust Feature-Sample Linear Discriminant Analysis for Neurodegenerative Brain Disorders Diagnosis', presents an extension of linear discriminant analysis with that explicit aim of gaining robustness against feature noise and outlier data. Based on the framework of [15], a criterion is derived, and yields a convex optimization, solved by an ALM method. Experiments are carried out on the PET dataset of ADNI, and the proposed method is shown to outperform alternatives.

Overall, I enjoyed reading the paper that puts together many interesting ideas. I learned quite a lot by reading it and the references.

However, there are a number of serious issues.

A prominent one is that the method makes use of the (unlabeled) test data X_{test}. This does not make sense in terms of medical diagnosis: you want to predict the outcome on a patient when you see his data for the first time, without retraining the model. I see 2 possibilities: - either the methods performs well without this rather problematic inclusion of test data, but this has to be shown. - or it does not, which drastically limits its applicability (yet does not invalidate the framework). In short, I want to see the results when the X_{test} data are not available at training time, and I consider this as a sine qua non requirement for acceptance.

A second aspect, is that the framework used by the authors is actually not so high-dimensional. One can thus wonder whether rather classical techniques such as regularized MCD, would perform well on this problem. See e.g. [MCD, RMCD]

[MCD] Rousseeuw, P. J., & Driessen, K. V. (1999). A fast algorithm for the minimum covariance determinant estimator. Technometrics, 41(3), 212-223.

[RMCD] Fritsch V, Varoquaux G, Thyreau B, Poline JB, Thirion B. Detecting outliers in high-dimensional neuroimaging datasets with robust covariance estimators. Med Image Anal. 2012 Oct;16(7):1359-70.

A third aspect is that the experimental setting is not realistic: p=100, n=500, noise-free model ! It should be easy to make it less advantageous. Note hat the above-mentioned MCD would work well in such a setting !

I tend to think that most of the case made by the authors about outliers is not serious: the authors rely on a quadratic model (LDA), while the logistic or hinge losses are robust by nature. I am not arguing against the results, but against the interpretation that this would come from higher robustness against outliers. Besides, taken from an application perspective, it is a bit surprising that outliers may be found in PET datasets, by nature of the physical measurement involved. Since my priors may be wrong I would like to know the proportion of outliers found in the dataset.

I have some hesitations regarding the results. It seems more than strange to me to see (R)LDA outperform SVMs by more than 15%. I have never seen this happen in any practical application. Is there a parameter tuning issue ?

Minor points ------------ A more sober title would be welcome

211-212: the sentence should be rephrased

272-278: hand-waving. The convergence seems to rely on a lot of tricks. The truth is that nobody knows how to tune mu.

282: the sentence should be rephrased
Summary: An interesting contribution, but I think that the claims are somewhat excessive and the experimental results are somewhat unexpected to me. More importantly, I explicitly urge the authors to clarify the importance of using the unlabeled test data at train time.

Submitted by Assigned_Reviewer_4

I found this paper mostly clear and well written, and results seem to indicate that the model perfroms better than competitors. Some small(?) remarks : * optimization seems a key aspect, but the robustness of results with small variations of the choices taken by the authors for some of the hyper parameters is not provided. There are difficulties in selecting hyper parameters that are not discussed * timing for computation and code availability are not discussed (how long did it take to run on the PPMI and ADNI?)

* RSF-LDA1 vs RSF-LDA2 ? * Why use so few of the ADNI data ?

* On the use of the ADNI data : I forgot to mention the issues with the use of these data given their data usage agreement - which is damaging for the scientific community.
Summary: The paper proposes an interesting model that finds a linear classifier robust to outlier observations and to noise in feature space. It is validated on simulations and two neuroimaging datasets, and shows better performance than standard techniques.

Author Feedback
Author rebuttal: We thank the reviewers for valuable comments. They collectively affirmed the originality of our work. We address their concerns here and in the paper.
Q1.ADNI data use agreement
In final version, we'll add "for ADNI" in the author list, include a footnote citing ADNI data use agreement, and acknowledge ADNI.
Q2.Results with no test-data involved (fully supervised setting)
We use test data (as unlabeled data during training) to form the geometry of sample space, and de-noise both train/test data, whilst building the model. If no test data is used, we can build the model, but only the train data (no test data) will be de-noised. We noted that the more samples used in de-noising, the better sample space is modeled. Accuracy for RFS-LDA2, learned using only train data, is:
PPMI:75.8%
AD/NC:86.3%
MCI/NC:84.5%

R1
1.Optimization
Objective (5) is convex, the optimization converges to the global optima (lines264-268). So, it's not sensitive to initial conditions.
2.Synthetic data
We construct 2 subspaces S1&S2 in R^100, with bases U1&U2. U1 in R^{100x100} is a random orthogonal matrix and U2=TU1 (T a random rotation matrix). 500 vectors are sampled from each subspace: Xi=UiQi, i=1,2 with Qi a 100x500 iid N(0,1). Noisy samples/features are drawn from N(0,1). Will clarify.
3.State-of-the-art for PD&AD
Neuroimaging data-driven PD diagnosis is scarce (this paper is among the firsts). State-of-the-art using ADNI includes [25,31,Yuan et al. KDD'12], which use multi-modal fusion or incomplete data classification. ADNI isn't complete for all modalities, so we used a portion of it with complete MRI+PET. These methods also use linear loss for feature selection/classification. Linear & nonlinear classifier performances are comparable (discussed in [25]).

R2
1.If method was probabilistic
If we use probabilistic k-NN after RFS-LDA mapping, we'll have probability outputs.
2.Mention of sensitivity & specificity
We reported AUC, which is related to sensitivity or TPR. The best sensitivity & specificity of RFS-LDA2 (10-fold CV) are respectively:
PPMI:0.88,0.81
AD/NC:0.98,0.85
MCI/NC:0.92,0.86
3.Small sample size (SSS) problem in Eq.1
Original LDA linearly maps X to y using covariance matrices. In case N< d, these matrices are probably rank-deficient. Directly minimizing Eq.1 (LS-LDA) avoids SSS problem (not using covariance matrices). Cited [9], but will clarify.
4.Why k-NN
LDA maximizes inter-class & minimizes intra-class variance. After it projects the samples to the output space, k-NN is used to determine class labels [9,13], as the projected space guarantees same-class samples to be closer to each other.
5.Notation
All defined in footnote1, page2.

R3
1.See Q2.
2.High-dimensional data and [RMCD]
[RMCD] proposes a nice/promising approach for outlier detection (will cite). It detects outlier samples, while our method deals with both sample-outliers and feature-noise. Further, we de-noise both train & test data. As for high dimensional data, the nature of solving LDA through the LS formulation guarantees a feasible mapping (see R2.3).
3.Non-realistic setting
In synthetic experiments(N=500,d=100), we add up to 200 noisy samples and 200 noisy features, to show the method's ability in noise and outlier suppression. [RMCD] works well in the case with 200 outliers, but might not be best when we have 200 noisy features. Moreover, PPMI data experiment shows results for N< d case (d=294,N=112).
4.Outlier in PET datasets
We used MRI for PPMI, and MRI+PET for ADNI. They contain deviant observations, due to acquisition, preprocessing artifacts or extreme inter-subject variability.
5.Portion of outliers
As in Eq.5, samples are weighted using \alpha. So, we actually don't discard outliers, but reduce their influence. Weights are modified iteratively by IRLS (not fixed). Roughly, we found about 1/4 of the samples in both PPMI & ADNI experiments were constantly getting low weights.
6.SVM results
SVM outperforms LDA, but is worse than RLDA/MC (tables1&2). Data is noisy and not all the features (brain ROIs) are useful. Methods with explicit noise modeling or feature selection perform better.

R4,R5,R6
1.Hyperparameters, mu params
We set the hyperparams with equations in lines 296-300, and set ratio params (\Lambdas) through a mild search in {0.01,0.1,1,10,100} (line300). \mu params (discussed in lines275-278) should increase smoothly, thus we initialize them (line219) and gradually increase using \rho=1.01.
2.Data split (lines297,311,362, we perform 10-fold CV), stopping strategy (all specified in algorithm lines 7,17).
3.Computation time
Baseline methods use either train samples or both train & test, some are inductive and some transductive. Direct run-time comparison is unfair. Instead, we provided theoretical computational complexity (lines269-278). Roughly, with MATLAB multi-core implementation and all parameters set, it takes 22min to build the model for PPMI and about 59min for ADNI.